# Perception of cattle owners towards risk of raw milk consumption for bovine tuberculosis transmission in Hosanna, Central Ethiopia: A community-based cross-sectional study

**Likawunt Samuel Asfaw**⬥*

Department of Clinical Nursing, Hosanna Health Sciences College, Hosanna, Central Ethiopia

* samuelliku@gmial.com

## Abstract

### Introduction

Zoonotic diseases account for more than 61% of human diseases. Raw milk is a major source of bovine tuberculosis (BTB) infection. However, there is a lack of comprehensive information on the community's perception of the risks associated with raw milk consumption for BTB transmission in Ethiopia. This study aimed to investigate the awareness of cattle farmers in Hosanna, southern Ethiopia, regarding the risk of bovine tuberculosis transmission through the consumption of raw milk.

### Methods

We conducted a community-based cross-sectional study among a randomly selected sample of households (n = 462) in Hosanna Town. We used pre-tested and structured questionnaires to collect data. The perception of the risk of bovine tuberculosis transmission due to raw milk consumption was assessed using the mean score of each outcome. Scoring above the mean on the four constructs of the Health Belief Model (HBM) is equivalent to having a high level of awareness of the risk of BTB transmission from raw milk consumption. 95% confidence intervals (CI) of the corresponding estimates were set to indicate significance.

### Results

The analysis results showed that 65.0% of the cattle farmers in the study area had a low awareness of the risk of BTB transmission from drinking raw milk. The perception of the risk of BTB transmission due to raw milk consumption was significantly lower in males (adjusted odds ratio (AOR): 2.6 CI 1.51, 4.68) and widowed (AOR: 3.7, CI 1.43, 9.92) participants.

**Data availability statement:** The datasets analyzed during the current study are available from the research and community services directorate of Hosanna Health Sciences College, and included the institutional address of the directorate, a focal person (Degefa Tadele, Tele: +251 0915671817, email: degefa04@gmail.com (Website of the college, https://www.hhsc.edu.et)

**Funding:** The author(s) received no specific funding for this work.

**Competing interests:** The authors have declared that no competing interests exist.

## Conclusion

In conclusion, the perception of the risk of raw milk consumption for BTB transmission is low in this study. Thus, it is worthwhile to include measures to enhance the perception of cattle owners toward the risk of raw milk consumption as a fundamental practice to control BTB transmission.

## Author summary

This study investigates the perceptions of cattle owners concerning the risk of bovine tuberculosis (BTB) transmission through the consumption of raw milk. Bovine tuberculosis, caused by Mycobacterium Bovis, is a zoonotic disease that poses a serious risk to the health of cattle and humans. Despite the known dangers, the consumption of raw milk remains a common practice in Ethiopia.

Through a series of structured interviews and surveys, we explored cattle owners' awareness and attitudes toward BTB and their milk consumption habits. The findings reveal gaps in awareness about the disease and its spread. Many cattle farmers are unaware that drinking raw milk can lead to BTB infection in humans. Furthermore, despite the risks, gender and marital status were significantly associated with raw milk consumption.

The study highlights the need for comprehensive, targeted education programs to increase awareness of BTB and promote safer milk consumption habits. By addressing these knowledge gaps, we can reduce the incidence of bovine TB transmission from cattle to humans and improve public health outcomes in affected communities.

## Introduction

Animals and humans live nearby and in the same physical habitat. In many developing nations, there is a deep connection between humans, animals, and the natural world. This link can potentially affect public health negatively and have far-reaching effects if not managed appropriately. The spread of infectious diseases is facilitated by an unavoidable relationship between people and animals [1]. Infections that spread among humans and animals are referred to as Zoonosis. BTB, Tapeworm, Ebola, Anthrax, and Brucellosis are a few zoonotic illnesses. Around the world, 60% of infectious diseases are caused by zoonosis[1.8].

Direct contact with the animal, vectors (such as fleas or ticks), or contaminated food or water can all result in the spread of the disease [2,3]. Studies have shown that drinking raw milk and/or consuming dairy products made from raw milk are the main sources of BTB infection in humans [4,5]. It is also spread by inhaling infectious

droplets or directly through broken skin. Cattle get the infection by inhaling infected droplets (aerosols) from human beings [4,5].

The study found that regular skin testing for known zoonotic infections, slaughtering exposed cattle, and limiting contact between cattle and wildlife can protect animals from BTB infection [6,7]. Pasteurization of milk and other dairy products containing milk can reduce the risk of human tuberculosis transmission [8]. Early diagnosis, slaughter of infected cattle, and avoidance of raw milk consumption are the three archetypal measures to control the risk of BTB infection in cattle and humans [9].

According to a World Health Organization (WHO) report, 9.6 million people contracted tuberculosis (TB) in 2014, with 25% of those cases occurring in Africa [8]. Apart from its extensive spread, tuberculosis is a severe illness that is known by its death rate, case fatality rate, and the psychological response in individuals with compromised immune systems (HIV/AIDS). Africa had the greatest recorded death rate from tuberculosis, accounting for over 1.5 million deaths [10]. Tuberculosis is the leading killer of children and young women aged 20–59. The 2015 Global Tuberculosis Report showed that 3.2 million young women developed active tuberculosis and 480,000 died. In a similar vein, 140,000 children worldwide died from tuberculosis in 2014, while an estimated 1.0 million youngsters contracted the disease [11].

Ethiopia is one of the highest TB-burdened countries with an estimated number of 220,000(261/100,000) New cases of active tuberculosis in 2010 [12]. It is the second leading cause of death in Ethiopia [12]. This devastating phenomenon provokes a psychosocial response due to its association with HIV/AIDS (human immunodeficiency virus/acquired immune deficiency syndrome) [13]. Moreover, TB due to M. bovis is found to cause severe forms of Extra-Pulmonary TB such as intestinal TB, Bone TB, and TB Lymphadenitis [13,14]. Zoonotic TB has similar clinical and pathologic features to TB caused by *Mycobacterium tuberculosis*. Advanced laboratory techniques involving bacterial culture of clinical specimens can be used to identify the pathogen that causes human tuberculosis. These factors affect the diagnosis and quantifying the proportion of human TB cases caused by M. bovis infection in Ethiopia [15].

It has been reported that 2.8% of TB infections were caused by BTB [16]. A systematic review evaluated the prevalence of Mycobacterium bovis and showed that an average of 17.0% (range: 16.7-31.4%) of human tuberculosis cases in Ethiopia were caused by Mycobacterium bovis [16]. Recent findings suggest that raw milk is a major source of tuberculosis infection. In Ethiopia, 81.8% of the population regularly consumes raw milk [17]. Factors that facilitate the spread of M. bovis through the consumption of raw milk remain prevalent in Ethiopia [18–21]. Despite the increased prevalence and susceptibility, the perceived risk of bovine TB from raw milk consumption is not well understood and is understudied in the study area. This study aimed to investigate the awareness of cattle farmers in Hosanna, southern Ethiopia, regarding the risk of BTB transmission through drinking raw milk.

## Theoretical framework of the study

We used the HBM constructs to operationalize the theoretical proposition of the study through the formulation of the specific objectives, the development of the study questionnaires, and the analysis of the results. HBM speculates that health-related decisions depend on the combined effects of: "one's perceptions of susceptibility" to a given health problem, 'perceived threat';' perceptions of barriers to adopting preventative measures'; perceptions of the benefits of taking specified health actions, and cues to action.

## Materials and methods

### Ethics statement

The Institutional Review Board of the Hosanna College of Health Sciences approved this study. To ensure confidentiality, personal identifying information, such as name, was removed from the data collection tool. Immediately before data collection, the purpose of the study was explained to all study participants, and informed verbal consent was obtained.

## Study design and population

We conducted a community-based cross-sectional study between March and April 2016 in the Hosanna community. Hosanna is the capital of the Central Regional State of Ethiopia, located approximately 232 km southwest of Addis Ababa. According to the 2007 [22]. The major ethnic groups in Hosanna Town include Hadiya, Kembata, and Gurage. In Hosanna, it is a common practice to use local breeds of cattle to produce milk. "Atakana" is the cultural diet of the Hadiya people that is prepared from milk, and its products are a favorite diet in Hosanna Town. In addition, raw milk is widely used by the community in the Town. We chose Hosanna for the current study because most residents of this city are at the greatest risk of BTB transmission.

## Sampling and data collection

The study included 5706 cattle-owning households identified through door-to-door visits using a screening questionnaire. During the screening visits, we contacted households that owned livestock and milk. The data collectors informed them and sought their consent to participate in the main study. Individuals who showed cooperation were assigned a code number (used as a research framework) and were asked to provide their contact address (telephone number), which could be used to contact them for data collection if selected. They were assured that the contact addresses they provided would be used for research purposes only. We consider these households as the source population. We determined the sample size using a single population proportion formula [23]. We assumed a Z statistic with a 95% CI (Z = 1.96) and at a precision level of 0.05, the expected proportion of cattle farmers aware of the risk of BTB transmission due to raw milk consumption would be 50% (P = 0.5).

We used the population proportion to size formula to allocate the sample size to five sub-administrative units of the town. Sac'h Duna administrative unit (total households = 6,995, households having livestock and access to milk ($N_1$ = 2104 households), sample size ($n_1$ = 154)). Addis Ketema administrative unit (total households 6,814, households having livestock and access to milk ($N_2$ = 2065), sample size ($n_2$ = 151)). Gofer Meda administrative unit (total households = 3211, households having livestock and access to milk ($N_3$ = 1522), sample size ($n_3$ = 112)). Bobicho Kale Hiwot Gospel administrative unit (total households 26, households having livestock and access to milk (N3 = 14), sample size ($n_4$ = 2)). Prison administrative unit (total households 26, households having livestock and access to milk (N5 2), sample size ($n_5$ = 1)) [22]. By considering the non-response rate (10%), we included (n = 462) sample households. We used a simple random sampling technique (using a random numbers table) to select households. From each of the selected households, one individual satisfying the inclusion criteria was selected by lottery method.

## Variables and measurement

The dependent variable of this study was the perception of cattle owners toward the risk of raw milk consumption for BTB transmission. Demographic variables were the independent variables in this study. We used a pre-tested structured questionnaire of the Amharic version for data collection. We developed the questionnaire by reviewing relevant literature [21] and conceptualizing items in constructs of HBM [Perceived susceptibility, perceived severity, and perceived benefits]. The questionnaire had four sections. The first section included questions on participants' age, gender, marital status, level of education, monthly income, religion, family size, and the highest level of education in the family.

The second section comprised nine questions eliciting knowledge on BTB, with response options of 'Yes' or 'No'. The overall knowledge of study participants was assessed using questions such as being able to mention causes of BTB, mode of transmission of BTB, identifying risk factors for BTB transmission, and preventive measures for BTB infection. A correct response scores one, and an incorrect response scores zero. We dichotomized the knowledge score to high (Score ≥ 5) and low (Score <5). The third section had 10 items inquiring about risk practices for BTB infection, including whether the study participants drink 'raw milk', with response options of either 'Yes' or 'No'. A response "Yes" to risk

practices to BTB infection scores one and "No" scores zero. Risk practices for BTB infection are dichotomized to high risk (Score ≥ 5) and low risk (Score <5). The fourth section consists of 20 questions relating to the health belief model constructs: perceived susceptibility (6 questions), perceived severity (5 questions), and perceived barriers (6 questions). We used the Likert scale to rate the responses of participants for the items in the health belief model constructs ("1=strongly disagree, 2=disagree, 3=Neutral, 4=agree, and 5=strongly agree"). We asked participants to rate their extent of agreement with the statements that best describe them. The higher the score, the higher the perception towards BTB that the subjects have. We dichotomized the responses to risk perception of BTB into High (score the median score for all constructs) and low (score < the median score for all constructs).

Reliability scale was undertaken for 63 items in all constructs, and internal consistency of the tool was estimated, and the tool was found to be internally consistent (Cronbach's Alpha = 0.83). Four Health extension workers and two Master of Public Health graduates who took training on basic principles of research ethics & data collection tools collected data and supervised the fieldwork, respectively.

## Statistical analysis

The collected data were processed using SPSS and STATA software. Computer-assisted data cleaning was performed through data exploration, simple frequency, consistency tabulation, and ordination techniques. To make any necessary revisions and further editing, outliers, missing values, and discrepancies were double-checked using the physical copies. We used summary metrics such as mean, median, and percentage to describe the data. We used proportion to describe Likert-scaled data. Meanwhile, Chi-square and Mann-Whitney U tests were used to analyze categorical and ordered Likert scale data. We used a logistic regression procedure to examine the effects of independent variables on BTB risk perception of raw milk consumption.

Candidate variables with a P-value less than 0.2 in univariable analysis were included in the multivariable regression model. We used the Backwards stepwise regression method to retain important predictors by removing the least significant variables at each step. P-value <0.05 is set to declare significance. We used SPSS 16 and STATA version 12.0 statistical software to process data. We used Mendeley's desktop to organize references. Findings were presented using the 95% CI of AOR with the corresponding P-value.

## Operational definition

**Perception towards the risk of raw milk consumption for BTB transmission**: Participants who scored above the median score for constructs in HBM were defined as having a high level of perception and those who scored less than or equal to the median for all constructs were defined as low level of level of perception towards risk of raw milk consumption for BTB transmission.

## Results

Four hundred sixty-two participants were contacted, five refused, and the data of nine participants were rejected for inconsistency. Data from 448 participants were valid and included in the analysis. Hadiya is the largest ethnic group in Hosanna (56.3%), followed by Kembata (14.3%). The majority of the participants were Protestant religious followers (48.2%), whereas 40.4% were orthodox Christian. The mean age of participants was 35. 33 ± 9.40 range (26–70) years. On average, 4.93 ± 1.63, range (2–11) people live together in single houses Table 1.

Of the 448 study participants who responded to the questions about their information on BTB infection, 26.1% had heard of BTB, but only 19.6% knew that it could be spread from animals to humans. Only 16.5% of respondents knew that consumption of infected milk could be a source of BTB infection in humans. A great majority (87.1%) of respondents had low knowledge of the basic premises of BTB infection Table 2.

**Table 1. Characteristics of participants responding to a survey on BTB in Hosanna, Ethiopia, 2016 (n = 448).**

| Characteristics | | Frequency | Percent |
|---|---|---|---|
| Sex | Male | 295 | 65.8 |
| | Female | 153 | 34.2 |
| Marital status | Married | 195 | 43.5 |
| | Single | 85 | 19.0 |
| | Widowed | 140 | 31.3 |
| | Divorced | 28 | 6.3 |
| | Others | 30 | 6.7 |
| Level of education | No education | 12 | 2.7 |
| | Primary | 250 | 55.8 |
| | Secondary | 92 | 20.5 |
| | College | 38 | 8.5 |
| | University | 56 | 12.5 |

**Table 2. Basic knowledge of participants responding to a survey in Hosanna, Ethiopia, 2016 (n = 448).**

| Knowledge questions | No (%) |
|---|---|
| Ever heard of BTB (n = 448) | |
| Yes | 117 (26.1) |
| No | 331(73.9) |
| TB is Zoonotic (n = 448) | |
| Yes | 88(19.6) |
| No | 360(80.4) |
| Drinking raw milk is the source of BTB in humans (n = 448) | |
| Yes | 74(16.5) |
| No | 374(83.5) |
| Sharing the same house with cattle is source of BTB (n = 448) | |
| Yes | 131(29.2) |
| No | 317(70.8) |
| Causes tuberculosis in Humans and cattle (n = 448) | |
| Bacteria | 131(29.2) |
| Others* | 317(70.8) |
| Overall knowledge score (n = 448) | |
| Good | 58(12.9) |
| Poor | 390(87.1) |

*: Cold air, hot climate, alcohol use, Smoking tobacco, shortage of food.

Age and sex did not have an association with knowledge of BTB infection. However, the marital status of respondents was significantly associated with their knowledge of BTB infection. Widowed respondents indicated a significantly lower knowledge of BTB infection (AOR: 6.0; CI 2.27-15.93) than currently married respondents. Respondents' responses concerning risk practices for BTB infection is depicted in Table 3.

The mean risk practice score was 3.91 ± (1.7), ranging from zero (minimum score) to seven (maximum score). The composite risk score was generated and the findings demonstrate, that over half of respondents (64.3%) consume raw

**Table 3. Risk practices for Bovine tuberculosis transmission among participants responding to a survey in Hosanna, Ethiopia, 2016 (n = 448).**

| Characteristics | No (%) |
|---|---|
| Consume raw milk (n = 448) | |
| Yes | 288(64.3) |
| No | 160(35.2) |
| Live together with cattle in the same house (n = 448) | |
| Yes | 199(44.6) |
| No | 249(55.6) |
| Consume raw meat (n = 448) | |
| Yes | 271(60.5) |
| No | 177(39.5) |
| Action when cattle got ill (n = 448) | |
| Nothing | 49(10.9) |
| Bring to veterinarian | 190(42.4) |
| Use traditional medicine | 173(38.6) |
| Others | 36(8.1) |
| Overall risk prevalence (n = 448) | |
| Low | 175(39.1) |
| High | 273(60.9) |

milk and (60.5%) of respondents consume raw meat. Nearly a quarter (44.6%) of respondents live together with cattle. The equated risk score indicated, 60.9% of respondents had a higher risk of BTB infection Table 3. The stated level of risk for BTB infection was significantly higher in males (AOR: 7.9; CI 3.28-19.26) than in females. Risk practice for BTB infection was not significantly associated with one's level of education. The association between marital status and risk practice for BTB infection was significantly higher for widowed compared to currently married respondents (AOR: 74.0; CI 8.4-625.9)

The majority (65.0%) of respondents had a low level of perception towards the risk of raw milk consumption for BTB transmission. The perceived susceptibility score ranges from eight (minimum) to 28 (maximum), with a median score (±SD) of 9.0, ±4.2, CI 8.0- 9.0. Most, 366 (81.7%) respondents indicated a low level of perceived susceptibility for BTB infection, Table 4 and Fig 1.

The median perceived severity score was 14.00, 6.9, CI 14.00-16.00. More than half, 238(53.2%) of respondents had a low level of perceived severity of BTB infection. The median perceived barrier score was 15.00 ± (5.7), CI14.46-16.56. The majority, 234(52.2%), CI 47.5- 57.4, of respondents have a high-level perceived barrier for BTB. Out of 448, 291(65.0%) CI 60.5- 69.4 respondents had a low level of risk perception for BTB infection, Table 5 and Fig 2.

We employed a chi-square test to examine the relation between the measured scores. Of the 231 respondents who had high knowledge of BTB, 66 (28.7%) demonstrated high risk for BTB infection. Respondents who had high knowledge of BTB infection indicated a lower likelihood of risk predisposition for BTB infection (AOR: 0.4; CI 0.27, 0.59) than those had low knowledge. We did not find a statistically significant difference between respondents' perceived susceptibility and knowledge of BTB infection. Although it was not statistically significant, high-risk groups were less likely to perceive susceptibility. In general, respondents having better knowledge about BTB infection were more likely to perceive BTB and its consequences Table 6.

The Hosmer-Lemeshow chi-squared statistic for the final model was 22.43 (p = 0.4), indicating that the model fitted the data well. Sex, marital status, and educational level of study participants indicated a statistically significant association with the perception of risk of raw milk consumption for BTB transmission. Males had significantly lower levels of

**Table 4. Perceived susceptibility of participants responding to a survey on bovine tuberculosis in Hosanna, Ethiopia, 2016 (n = 448).**

| Response questions | Strongly disagree: No (%) | Disagree: No (%) | Neutral: No (%) | Agree: No (%) | Strongly agree: No (%) |
|---|---|---|---|---|---|
| Rate the chance of contracting BTB (n = 448) | 226(50.4) | 184(41.1) | 20(4.5) | 8(1.8) | 10(2.2) |
| There is an increased risk of contracting BTB when drinking raw milk (n = 448) | 250(55.8) | 96(21.4) | 17(3.8) | 66(14.7) | 19(4.2) |
| Risk of contracting BTB when eat raw meat (n = 448) | 265(59.2) | 107(23.9) | 50(11.2) | 20(4.5) | 6(1.3) |
| Risk of contracting BTB when living together with cattle | 309(69.0) | 47(10.5) | 82(18.3) | 5(1.1) | 5(1.1) |

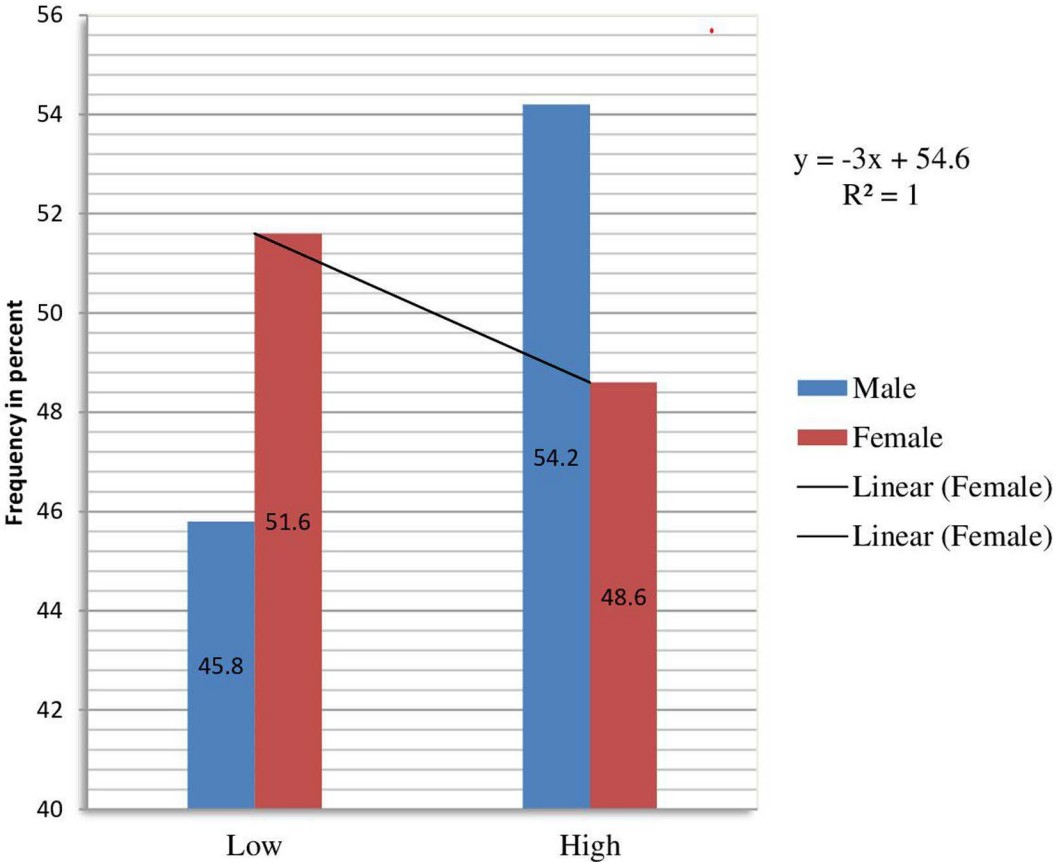

$$y = -3x + 54.6$$
$$R^2 = 1$$

**Fig 1. Perceived barriers to preventive practices for BTB transmission by sex.**

**Table 5. Perceived severity of bovine tuberculosis infection of participants responding to a survey in Hosanna, Ethiopia, 2016 (n = 448).**

| Response questions | Strongly disagree: n (%) | Disagree: n (%) | Neutral: n (%) | Agree: n (%) | Strongly agree: n (%) |
|---|---|---|---|---|---|
| BTB affects work | 75(16.8) | 54(12.1) | 129(28.9) | 57(12.8) | 132(29.5) |
| BTB keeps me ill | 118(26.4) | 47(10.5) | 98(21.9) | 64(14.3) | 120(26.8) |
| BTB causes death | 65(14.5) | 121(27.1) | 102(22.8) | 36(8.1) | 123(27.5) |
| BTB damages the organ | 104(23.3) | 123(27.5) | 27(6.0) | 65(14.5) | 128(28.6) |

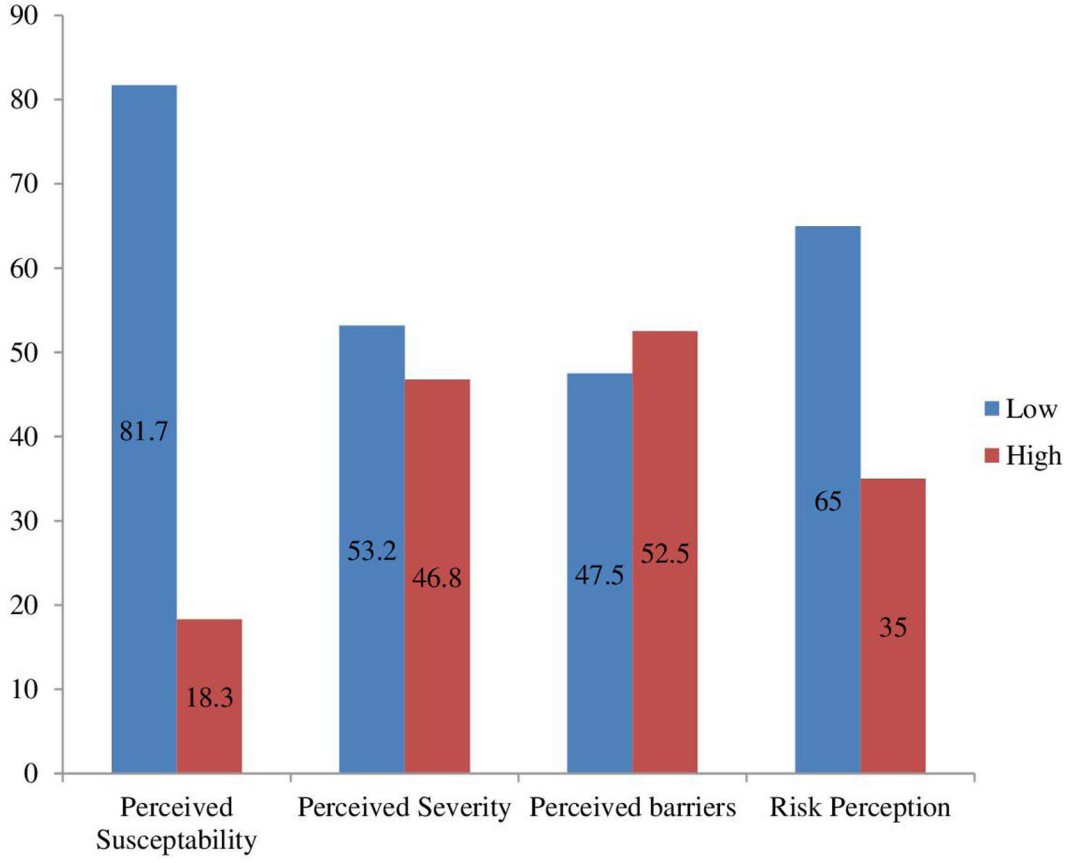

**Fig 2. The distribution of risk perception of raw milk consumption.**

**Table 6. Chi-square test findings of BTB risk practice of raw milk consumption among participants responding to a survey on bovine tuberculosis in Hosanna, Ethiopia, 2016 (n = 448).**

| Characteristics | High risk for BTB OR (CI) | High perceived susceptibility OR (CI) | High perceived severity OR (CI) |
|---|---|---|---|
| High knowledge | 0.4 (0.27,0.59) | 1.2(0.78, 2.04) | 3.3 (2.20, 5.00) |
| High risk for BTB | | 0.09 (0.03, 0.21) | 0.27 (0.18, 0.40) |
| High perceived susceptibility | | | 0.03(0.008, 0.14) |

perception towards the risk of raw milk consumption for BTB transmission (AOR: 2.6 (1.51, 4.68) than females. Likewise, widowed participants had significantly lower levels of perception towards the risk of raw milk consumption for BTB transmission (AOR: 3.7 (1.43, 9.92) than currently married participants Table 7.

## Discussion

In this study, we have assessed the perception of cattle owners on the risk of raw milk consumption to BTB transmission among people of different socioeconomic strata in Hosanna. We used the health belief model to explicitly explain risk perception and identify factors determining high-risk practices for BTB transmission. Understanding risk perception leads

**Table 7. Regression findings of perception towards risk of raw milk consumption for BTB transmission among study participants in Hosanna, Ethiopia, 2016 (n = 448).**

| Characteristics | Perception towards the risk of raw milk consumption for BTB transmission | | AOR†(CI) | P-value |
|---|---|---|---|---|
| | Low | High | | |
| **Sex** | | | | |
| Male | 181(62.2) | 114(72.6) | 2.6 (1.51,4.68) | 0.001* |
| Female | 110(37.8) | 43(27.4) | 1 | |
| **Marital status** | | | | |
| Currently married | 114(39.2) | 81(51.6) | 1 | |
| Single | 58(19.9) | 27(17.2) | 1.02(0.54,1.94) | 0.93 |
| Divorced | 109(37.5) | 31(19.7) | 0.76(0.40,1.42) | 0.06 |
| Widowed | 10(3.4) | 18(11.5) | 3.7 (1.43,9.92) | 0.001* |
| **Education level** | | | | |
| No formal education | 196(67.4) | 51(32.5) | 9.7(5.35,17.71) | 0.001* |
| Primary education | 29(10.0) | 66(42.0) | 4.1 (1.91,8.6) | 0.001* |
| Secondary education | 16(5.5) | 22(24.0) | 0.63(0.28,1.39) | 0.2 |
| College graduate | 46(15.8) | 10(6.4) | 5.5(1.48,20.4) | 0.01 |
| University graduate | 4(1.4) | 8(5.1) | 1 | |

†-adjusted for sex, marital status, education level, knowledge score and age * P-value < 0.05, 1-Reference.

to a better understanding of risk decision-making and risk behavior. The study has established the low prevalence of risk perception of raw milk consumption for BTB. We also identified low knowledge of the basic premises of BTB infection and a high prevalence of risk practices for BTB transmission.

In the current study, only 9.8% of respondents had good knowledge related to BTB and its means of transmission. This finding is similar to the results of a study from Northern Ethiopia. They concluded that 9.3% of the community in Woldiya town had good knowledge related to BTB and its means of transmission [19]. However, the finding of the current study is lower than reported for small-scale dairy farms in Adama town [24]. Similarly, a study from Arsi in Eastern Ethiopia [25] identified that 77% of respondents have good knowledge of BTB, which is far higher than the current finding. Despite a large portion of study participants having heard of BTB, only 21.7% knew TB was transmitted through raw milk in the current study. This finding is in line with the study reported for the community in Dilla, Northern Ethiopia [26]. However, the current finding is lower than the previous study that reported for Northeast Ethiopia [19]. Consistencies in reports imply a low level of knowledge on basic premises of BTB infection among the community in Hosanna.

Unlike to previous report [20] majority (56.9%) of the community in the current study engaged in known risk practices for BTB transmission. We have found that a large portion (64.3%) of respondents consume raw milk. Our findings show similar patterns to data on raw milk consumption in Gondar (81.8%) [17], Arisi (55.4%) [25], and Dilla (80%) [26].

The current study showed that only 18.3% of respondents perceived susceptibility to BTB infection and 46.8% perceived severity of BTB infection. This is the first study examining the perceived susceptibility and severity of BTB infections. Protection motivation theory states that perceived threats can influence the intention to adopt preventive measures. As a perceived threat, here is the combined effect of both susceptibility and severity. Low levels of these driving forces in the current study strongly suggest the need for optimal approaches known to enhance the perception of the community on susceptibility and severity of BTB infection.

We identified that 65.0% of respondents had a low level of perception towards the risk of raw milk consumption for BTB transmission. The Level of perception in the current study is lower than reported for the Itang community in Gambella (Western Ethiopia), 50.9% [21]. Nevertheless, the findings of the current study are in agreement with the study reported for dairy farm owners in Nigeria [27]. Methodological and time variations could more likely explain the observed difference.

The previous study assessed perception by directly forwarding a few questions about attitude and perception. However, the current study used items in the health belief model.

The level of knowledge of the community and the biological nature of the disease could likely explain the reduced level of risk perception for BTB infection. The majority of the participants had a low level of knowledge and engaged in higher-risk practices for BTB transmission. The concept of "Optimism bias" could more likely explain the observed difference. According to this concept, individuals tend to believe BTB is transmitted from infected people rather than from consuming raw milk.

When people do not develop the disease within a short period of exposure, they subsequently develop" risk tolerance" that underestimates the risk of susceptibility and risk of severity of the disease [28]. The community in Ethiopia believes that BTB infection is transmitted only from people having active TB. Correspondingly, BTB is a chronic disease that which the disease signs and symptoms appear after at least two weeks of infection. Accordingly, the low level of knowledge and increased risk (regular raw milk consumption) without immediate signs of disease were found to pose increased risk tolerance that could finally result in reduced risk perception.

The result of the current study indicated that sex, marital status and level of education of participants were important predictors of perception towards risk of raw milk consumption for BTB transmission. Females have a good level of risk perception, which is consistent with a previous study reported for the Itang community in Gambella [21]. Observations claimed that women value their health more than men which could likely elucidate the current finding.

In the present study, it was found that the perception of the risk of raw milk consumption for BTB transmission was significantly lower among widowed than currently married participants. This finding is inconsistent with previous studies in Nigeria [29]. The difference could be due to geographical and cultural variations between study participants. The previous study was done in Nigeria among dairy farm communities that have different geographical and cultural backgrounds. More often widowed people lack support and give lower value to their health. This could more likely explain the observed difference; however, the association between perception towards the risk of raw milk consumption for BTB transmission and marital status warrants further study. The macro-level factors, such as Culture and religion, were not significantly associated with perception in this study. Lower risk perception regardless of the unacceptable reason is an implication of increased risk tolerance and encourages higher risk behavior for BTB transmission [28].

The present study has some relevant limitations that impede its power. One of the limitations of this study is related to the cross-sectional study design, in which the temporal relationships between the outcome and predictor variables cannot be established. Moreover, the sample was limited to a single population, which can limit the power of the study. We recommend an exhaustive exploration of the factors associated with the perception of the risk of raw milk consumption for BTB transmission among different segments of the community in Ethiopia. Despite the stated limitations, this study can serve as a critical input for health programmers aiming to tackle the identified gaps.

## Conclusion

The perception of the risk of raw milk consumption for BTB transmission is low in the study area. Sex, marital status, and level of education showed significant association with a low level of perception towards the risk of raw milk consumption for BTB transmission. An increased risk practice for BTB infection and a low level of perception towards the risk of raw milk consumption for BTB transmission suggest the need for an integrated intervention to mitigate the risk of BTB transmission.

## Acknowledgments

I would like to thank Jimma University School of Graduate Studies, "One Health Central, and the East Africa (OHCEA) project. I would also like to thank the Hosanna Health Sciences College research and community service directorate. The author is also grateful to Hosanna town residents, data collectors, and the Hosanna town health office for their

cooperation during the entire process of data collection. My thanks also go to all participants for their willingness to participate. My special thanks also go to the data collectors and supervisors for their unreserved effort to collect quality data.

## Author contributions

**Conceptualization:** Likawunt Samuel Asfaw.

**Data curation:** Likawunt Samuel Asfaw.

**Formal analysis:** Likawunt Samuel Asfaw.

**Investigation:** Likawunt Samuel Asfaw.

**Validation:** Likawunt Samuel Asfaw.

**Writing – original draft:** Likawunt Samuel Asfaw.

**Writing – review & editing:** Likawunt Samuel Asfaw.

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
