## [Decision Letter · Decision Letter 0]

10 Jul 2025

Dear Dr. Asfaw,

We are pleased to inform you that your manuscript 'Perception of Cattle Owners towards Risk of Raw Milk Consumption for Bovine Tuberculosis Transmission in Hosanna, Central Ethiopia: Community-based Cross-sectional Study' has been provisionally accepted for publication in PLOS Neglected Tropical Diseases.

Also, the reviewer's minor comments need to be addressed. 

Best regards,

Lawrence Mugisha, PhD

Academic Editor

Qu Cheng

Section Editor

Shaden Kamhawi

co-Editor-in-Chief

Paul Brindley

co-Editor-in-Chief

Reviewer's Responses to Questions

**Key Review Criteria Required for Acceptance?**

**Methods**

-Are the objectives of the study clearly articulated with a clear testable hypothesis stated?

-Is the study design appropriate to address the stated objectives?

-Is the population clearly described and appropriate for the hypothesis being tested?

-Is the sample size sufficient to ensure adequate power to address the hypothesis being tested?

-Were correct statistical analysis used to support conclusions?

-Are there concerns about ethical or regulatory requirements being met?

Reviewer #1: (No Response)

**Results**

-Does the analysis presented match the analysis plan?

-Are the results clearly and completely presented?

-Are the figures (Tables, Images) of sufficient quality for clarity?

Reviewer #1: (No Response)

**Conclusions**

-Are the conclusions supported by the data presented?

-Are the limitations of analysis clearly described?

-Do the authors discuss how these data can be helpful to advance our understanding of the topic under study?

-Is public health relevance addressed?

Reviewer #1: (No Response)

**Editorial and Data Presentation Modifications?**

Reviewer #1: (No Response)

**Summary and General Comments**

Reviewer #1: General Comments

This manuscript is a great contribution to the understanding of risk of raw milk consumption and surveillance of bovine tuberculosis. The manuscript was well articulated but still requires little grammar checks.

Specific Comments

L100: Italize all bacterial names e.g. Mycobacterium tuberculosis

L135: How did you arrived at this sample size (5,706)?

Provide separate ‘Conclusion’ section for the manuscript.

PLOS authors have the option to publish the peer review history of their article (what does this mean? ). If published, this will include your full peer review and any attached files.

**Do you want your identity to be public for this peer review?** For information about this choice, including consent withdrawal, please see our Privacy Policy .

Reviewer #1: **Yes: ** Nma Bida ALHAJI

---

## [Editor Report · Acceptance letter]

Dear Dr. Asfaw,

We are delighted to inform you that your manuscript, " 

Perception of Cattle Owners towards Risk of Raw Milk Consumption for Bovine Tuberculosis Transmission in Hosanna, Central Ethiopia: Community-based Cross-sectional Study," has been formally accepted for publication in PLOS Neglected Tropical Diseases.

Best regards,

Shaden Kamhawi

co-Editor-in-Chief

Paul Brindley

co-Editor-in-Chief
